# Dietary Behavior and Risk of Orthorexia in Women with Celiac Disease

**DOI:** 10.3390/nu14040904

**Published:** 2022-02-21

**Authors:** Karolina Kujawowicz, Iwona Mirończuk-Chodakowska, Anna Maria Witkowska

**Affiliations:** Department of Food Biotechnology, Faculty of Health Sciences, Medical University of Bialystok, Szpitalna 37, 15-295 Bialystok, Poland; iwona.mironczuk-chodakowska@umb.edu.pl (I.M.-C.); anna.witkowska@umb.edu.pl (A.M.W.)

**Keywords:** celiac disease, orthorexia, eating disorders, ORTO-15, nutrition

## Abstract

Evidence points to a link between celiac disease and eating disorders. Although with the current limited knowledge, orthorexia cannot be formally recognized as an eating disorder, some features are similar. This study is the first to examine individuals with celiac disease in terms of the prevalence of risk of orthorexia. Participants were 123 females diagnosed with celiac disease. The standardized ORTO-15 questionnaire was used to assess the risk of orthorexia. In this study, eating habits and physical activity were assessed. The effect of celiac disease on diet was self-assessed on a 5-point scale. Taking a score of 40 on the ORTO-15 test as the cut-off point, a risk of orthorexia was found in 71% of individuals with celiac disease, but only in 32% when the cut-off point was set at 35. There was a positive correlation between age and ORTO-15 test scores (rho = 0.30). In the group with orthorexia risk, meals were more often self-prepared (94%) compared to those without risk of orthorexia (78%) (*p* = 0.006). Individuals at risk for orthorexia were less likely to pay attention to the caloric content of food (46%) relative to those without risk of orthorexia (69%) (*p* = 0.001). For 64% of those at risk for orthorexia vs. 8% without risk had the thought of food that worried them (*p* = 0.001). Given the survey instrument for assessing the prevalence of orthorexia and the overlap between eating behaviors in celiac disease and orthorexia, the prevalence of orthorexia in celiac disease cannot be clearly established. Therefore, future research should focus on using other research tools to confirm the presence of orthorexia in celiac disease.

## 1. Introduction

Celiac disease is an autoimmune disease characterized by malabsorption resulting from chronic inflammation of the small intestinal mucosa caused by gluten ingested in the diet [1]. It affects people of all ages around the world with an estimated prevalence of 1% in the population of Europe and the USA, but an increasing trend is observed in many countries [2]. Celiac disease is primarily diagnosed in women, in part because of its higher prevalence, but also because women are more likely than men to undergo diagnostic testing [3,4]. In addition to pathophysiological background, common symptoms in untreated celiac disease are anxiety and depression [5]. The primary therapy in patients diagnosed with celiac disease is strict adherence to a gluten-free diet throughout life [6], which may lead to psychological distress during transition into a gluten-free diet [7]. Recent research suggests a significant role of psychological distress as a risk factor for disordered eating in celiac disease [7].

Orthorexia nervosa is an excessive preoccupation with eating healthy foods, which consequently may potentially lead to significant dietary restrictions and, in extreme cases, to abandon food consumption altogether [8]. The average prevalence rate of orthorexia is 6.9% for the general population and 35–57.8% for high-risk groups and depends on diagnostic cut-off criteria, tool used, cross-cultural, geographic and sex differences [9,10]. Tendencies toward orthorexia are comparable between genders, although it is slightly more pronounced in women [11]. The diagnostic criteria for orthorexia have not yet been standardized [12], and orthorexia has not been formally recognized as an eating disorder [13], although there is considerable overlap between the symptoms of orthorexia and eating disorders [14]. Dunn and Bratmann [15] described criteria which addressed the presence of obsessive focus on healthy eating and clinically significant compulsive behaviors and excessive mental focus on healthy eating resulting in adverse nutritional status. Moroze et al. [16] proposed criteria relating to excessive focus on the quality and composition of meals and relating to the exclusion of the influence of other diseases, food allergies, religious beliefs or diseases requiring a specialized diet. The well-known and commonly used tool to assess the risk of orthorexia is the ORTO-15 questionnaire developed by Donini et al. [17]. 

While people with celiac disease must exclude gluten from foods for health reasons, people with orthorexia voluntarily pursue healthy eating through a restrictive diet, focusing on food preparation and analyzing the sources of the foods they eat [18]. People with orthorexia exhibit obsessive behaviors related to maintaining and improving their physical health and well-being [19,20]. Excessive focusing on elimination, initially of small groups of products, especially unhealthy, i.e., fried, grilled and rich in simple sugars, and later of whole groups of products, leads in consequence to deficiencies of calories, nutrients, vitamins and minerals [21]. In addition, orthorexia can lead to real difficulties both physical and psychological, such as gastrointestinal problems, epigastric pain, anemia, osteopenia, social isolation, feelings of guilt and lack of self-acceptance after consuming unadvisable food [21]. 

Focusing on diet due to medical reasons is recognized as a potential risk factor for eating disorders and orthorexia [22,23]. Based on existing scientific evidence, our study’s search for an association between the risk of orthorexia and celiac disease in adults seems reasonable. There are no studies in the literature on the association of celiac disease with orthorexia, but there are reports of an association between celiac disease and the occurrence of eating disorders [22]. Passannanti et al. [24] found that celiac disease itself whether or not co-occurring gastrointestinal symptoms or psychological factors may contribute to pathological eating behaviors in adults with celiac disease. It has been observed that eating disorders may be more common in young females with celiac disease than in men, both healthy and with celiac disease. The increased frequency of eating disorders may be related to the obsessive consumption of some types of food [25], as for example intake of added sugar has been shown to be higher in people with celiac disease than in healthy individuals [26].

The number of celiac diagnoses is increasing, and this is probably due to a real increase in incidence rather than increased awareness and detection [27]. With the increasing prevalence of celiac disease, there may be an increasing risk of orthorexia, which is still a newly recognized eating style [25], that has much in common with eating disorders [14]. The purpose of this study was to assess the risk of orthorexia and to evaluate the eating behaviors of individuals with celiac disease. The study is the first to examine the association of celiac disease with orthorexia in conjunction with specific eating behaviors.

## 2. Materials and Methods

### 2.1. Study Group

The study group was recruited in 2018–2019 from a convenience sample, offering no incentives to avoid self-selection bias, by the official website of the celiac disease association, as well as through social media for people with celiac disease. The inclusion criteria adopted in this study were: age over 18 years, diagnosis of celiac disease and consent to voluntarily participate in the study. Applicants were verified in two stages. In the first step, it was determined via email communication whether they met the inclusion criteria for the study prior to completing the survey. In the second stage, age and presence of celiac disease were verified in the survey. 

The sample size was calculated based on the prevalence of celiac disease in the Polish population, which is estimated at 1%, with a 95% confidence interval and a margin of error of 1.71%. Out of 134, a total of 130 applicants were positively verified for inclusion criteria. All individuals were supervised by a gastroenterology, internal diseases and food allergy clinic. At the time of the study, no patient had been hospitalized for increased symptoms of celiac disease. Celiac disease was clinically confirmed in all participants, who were asked for medical records with a clinical diagnosis of celiac disease. The basis of diagnosis in subjects was the presence of (1) positive serological tests for at least one of the following antibodies: the anti-tissue transglutaminase (tTG), anti-smooth muscle endomysial (EmA) antibodies and (2) histopathological findings on small bowel biopsy with a Marsh score of at least 3a. Due to the anonymous nature of the study, medical records were not included in this paper, except for a summary of the number of diagnoses in the respondents. 

Despite the relatively lengthy recruitment process, a limited number of men participated in the study. Therefore, to avoid bias of inappropriate gender ratio, the study group was eventually restricted to women. Thus, the final study group consisted of 123 females verified for inclusion criteria. A flowchart of the study participants is shown in Figure 1.

In the study, the duration of celiac disease was classified in two intervals: (1) up to 3 years (newly diagnosed celiac disease) and (2) more than 3 years (long-term duration). The largest group of patients were those who have been diagnosed with celiac disease for more than 3 years (*n* = 94), followed by newly diagnosed patients (*n* = 29). Demographic information was collected that included age, gender, education and work situation. The health history included methods of diagnosis, duration of celiac disease and presence of comorbidities. Weight status was assessed with anthropometric measurements to determine BMI for which the 2019 WHO standards were adopted [28]. Eating behavior was assessed with a questionnaire developed for this study. 

The study was approved by the Local Bioethics Committee.

### 2.2. Eating Habits and Physical Activity

Dietary habits were examined using a dietary survey questionnaire. The frequency of eating meals per day, the difficulty of eating meals out, the way of preparing gluten-free meals, paying attention to the calorific value of food and the composition of food products, as well as the influence of celiac disease on the way of eating were analyzed on a 5-point intensity scale (1-does not influence at all or to a small extent, 5-influences very much).

Physical activity and frequency of physical activity was analyzed using the survey questionnaire. The questions covered several types of physical activity such as aerobic (fitness), swimming, running, strength training gym, walking, cycling and yoga. Frequency of physical activity was based on frequency per week: 1 time per week, 2–3 times per week, 4 or more per week, daily or no physical activity.

### 2.3. Questionnaire ORTO-15

The validated Polish version of the ORTO-15 self-report questionnaire was used in this study [29,30]. The ORTO-15 questionnaire has not been validated to assess orthorexia in a population with celiac disease, therefore, to understand internal consistency in this population, the reliability of the questionnaire was assessed using Cronbach’s alpha. This questionnaire was based on the original questionnaire ORTO-15 by Donini et al. [14]. Cronbach’s alpha coefficient in previous Polish studies ranged from 0.64 to 0.9 [29,30]. In the ORTO-15 test each question is answered by the subject on a four-point Likert scale: always, often, rarely or never, which are originally assigned 1, 2, 3 or 4 points, respectively. Because our study was the first to use the ORTO-15 questionnaire in celiac population, therefore, we retained the original cut-off criteria of 40 points [17] to compare our results with other work. In the original ORTO-15 questionnaire, scoring lower than 40 points indicates a propensity for orthorexia. The results obtained with the ORTO-15 questionnaire were interpreted using the reference recommendations of its authors by calculating the total score assigned to individual responses. The sum of points obtained by the respondent (theoretical range: 15–60) constitutes the so-called orthorexia risk index (ORI).

### 2.4. Statistical Analysis

The results obtained were statistically analyzed using Statistica 13.3 StatSoft software. Results with a significance level of *p* < 0.05 were considered significant.

Chi-square test of concordance, Spearman’s correlation and alpha-Cronbach’s coefficient for reliability analysis of psychological tests were used for data analysis and intergroup comparisons.

The ORTO-15 questionnaire was subjected to reliability analysis to confirm the consistency of the test. The reliability analysis of the ORTO-15 questionnaire included 15 questions and responses to the questions provided by the research participants. The reliability of the questionnaire was analyzed by examining the statistical properties of the test items and the relationship of the test items to the overall test score. An alpha Cronbach’s coefficient between 0.70 and 0.90 was taken as a satisfactory level of consistency of the questionnaire as recommended [31,32].

## 3. Results

### 3.1. Characteristics of the Study Participants

Table 1 shows the general characteristics of the study participants. The study group consisted of 100% females (*n* = 123). The median age was 34 years, and the median BMI was 21.4 kg/m^2^. The majority of subjects had tertiary education (80.5%). Permanent employment declared 74.8% of respondents. Normal body weight was observed in the majority of the group, 73 %. The largest group of people, about 76%, were patients for at least 3 years. Those newly diagnosed up to 3 years represented 24% of the study population. The most common comorbidities were: lactose intolerance (in 23% of individuals) and Hashimoto’s disease (in 22% of individuals). Strict adherence to a gluten-free diet was declared by 100% of respondents. In the subjective assessment of their health status, as many as 87% of the subjects rated that their health was better after following a gluten-free diet.

### 3.2. Reliability Assessment of the ORTO-15 Questionnaire 

Raw Cronbach’s α coefficient in this study was 0.67 and the standardized coefficient was α = 0.66. Table 2 shows the mean scores, the correlations of each question with the overall score, as well as the values of the raw and standardized Cronbach’s α coefficients, after the removal of a question from the questionnaire, if any. The highest correlation coefficients were recorded for the influence of mood on eating behavior (rho = 0.58), worries about eating for the past 3 months (rho = 0.55) and for more than 3 h a day (rho = 0.55) and the belief of increased self-esteem due to eating only healthy foods (rho = 0.51).

Table 3 shows the reliability analysis for a score that includes 15 questions (ORTO-15) and excludes question 8. These questions decreased the internal consistency of the ORTO-15 questionnaire, because the raw Cronbach’s alpha coefficients for these questions deviated from the sum of the standardized Cronbach’s alpha coefficient values for the other questions in the questionnaire, for which it was 0.66. The Cronbach’s alpha coefficient and, thus, the internal consistency of the questionnaire was higher after excluding question 8 and was 0.70. The mean score of the ORTO-15 questionnaire was 37.73 ± 5.45 points with a range of 23–53.

### 3.3. Prevalence of Orthorexia in Celiac Participants

Two cut-off points were used to assess the presence of orthorexia: a score of 40 on the original ORTO-15 questionnaire (17) and an arbitrarily adopted cut-off point of 35 points in other studies (37, 38). According to these cut-off points, the number of adults with celiac disease showing risk of orthorexia is 71% at the 40 cut-off point and 32% at the 35 cut-off point (Table 4).

As shown in Figure 2 the total ORTO-15 test score was significantly different between individuals with and without risk for orthorexia (*p* = 0.0001).

### 3.4. The Relationship between the Age of Study Participants and the ORTO-15 Questionnaire Total Score

Relationship between age of participants and the ORTO-15 scoring was shown in Figure 3. There was a positive correlation (rho = 0.30, *p* = 0.001) found in the Spearman’s correlation test. The interpretation of this result is that the higher the total ORTO-15 score, the lower the risk of orthorexia.

### 3.5. Eating Habits and Behaviors and the Incidence of Orthorexia Risk

This study found that both groups of participants, these at risk and those without the risk of orthorexia, were most likely to eat 4 meals per day, 38% and 67% participants of each group, respectively (Table 5). There was no predominant category of the number of meals consumed per day in people at risk of orthorexia, but it was typical for the people without risk to consume four meals per day. In addition, those, both, without risk of orthorexia (78%) and at risk of orthorexia (94%) were more likely to choose to cook their own meals than to have a family member prepare their meals. Great difficulty in eating out was experienced by both those at risk of orthorexia 78% and those without risk of orthorexia 92%, these results were close to statistical significance. Analysis of eating behaviors showed that the group of subjects at risk for orthorexia were more likely to pay attention to the calorie content and most of them (87%) paid attention to the composition of gluten-free products, but these results were not statistically significant. Physical activity among the respondents was also investigated (Table 5). 70% of those at risk for orthorexia and 81% of those without risk did some physical activity. Individuals with risk for orthorexia were significantly less likely to run (17%) and to exercise aerobic (31%) than those without the risk of orthorexia.

### 3.6. Differences between the Risk Group of Orthorexia and Non-Risk Participants in ORTO-15 Questionnaire

Chi-square test was used to assess the relationship of individual responses to the ORTO-15 questionnaire with the prevalence of orthorexia risk (Table 6). 

Participants at risk for orthorexia were more likely to experience anxiety related to thoughts about food for the last 3 months (64% vs. 8%) and more than 3 h a day (22% vs. 0%) than people with celiac disease without orthorexia, found in the ORTO-15 test. People at risk of orthorexia slightly more often felt guilty when committing dietary transgression, 42%, compared to 39% of those without risk of orthorexia. Individuals at risk of orthorexia relative to those without risk of orthorexia were less likely to pay attention to the caloric content of food (46% vs. 69%), were less likely to consider taste more important than food quality (49% vs. 72%), and were more emotionally stable with respect to food choices: less likely to feel confused when entering a store (73% vs. 94%) and less likely to have their mood influence their eating behavior (20% vs. 80%). In addition, those at risk for orthorexia were more likely to believe that consuming healthy food could improve their appearance 83% compared to those without orthorexia risk 59%. Participants at risk orthorexia risk were willing to spend more money than those without the risk of orthorexia to buy healthier foods.

### 3.7. Self-Assessed Impact of Celiac Disease on Diet

Figure 4 shows the self-assessed impact of celiac disease on participants’ diet using a five-point Likert scale according to orthorexia risk status. The percentage of subjects gradually increased as the impact of celiac disease on diet increased in both at-risk and non-at-risk celiac participants. Among those with no risk of orthorexia, as many as 75% rated the impact of celiac disease as 4–5 in the 5-point scale. Similar results were found in the group with risk of orthorexia, 74%. The results were not statistically significant (*p* = 0.352).

## 4. Discussion

In our study, the ORTO-15 questionnaire was tested for reliability using Cronbach’s alpha test. The results of our and other studies [33,34] indicate differences in the reliability of the ORTO-15 questionnaire.

Our study suggests that there may be disturbing eating behavior in females with celiac disease and this topic should be further investigated. The study by Sutherley et al. [35] indicated eating disorders in celiac disease. To date, several tests have typically been used to screen for orthorexia in various populations, but not in celiac disease. Brytek-Matera et al. [36] used Düsseldorf Orthorexia Scale and the Eating Habits Questionnaire to investigate the prevalence of orthorexia in Polish healthy participants. The prevalence rate of orthorexia in that study was 2.6%, while in Lebanese healthy adults the prevalence rate was 8.4% [37]. In another study, the prevalence of orthorexia nervosa among dietetics and nutrition students in Jordan was 72% with a cut-off score of 40 and 31.8% with a cut-off score of 35 of the ORTO-15 questionnaire [38]. Dunn et al. [39] examined the prevalence of orthorexia among US students using ORTO-15 questionnaire 71.2% score with the original cut-off score of and 22.1% when a cut-off score of 35 was applied. Our results were similar to previous studies.

Our study showed that the risk of orthorexia decreases with age, and is highest among younger individuals. Several studies have also shown that orthorexia is more common among younger than older adults [40,41]. Furthermore, with respect to celiac disease, Bongiovanni et al. [42] found that the stress of having to follow a gluten-free diet affects mainly young people and those with the disease lasting for less than four years. Changing eating habits is a long, difficult process and can contribute to stress in newly diagnosed individuals.

As the number of men with celiac disease recruited for this study was small, they were not taken into account during data analysis. Therefore, in this study, females constituted a total percentage of the participants. As studies indicate, celiac disease is more frequently diagnosed in women. Gender differences in adults with celiac disease were shown by West et al. [43], who found that the prevalence of celiac disease in females was significantly higher, especially in Finland, the Netherlands, Denmark and the UK.

In recent years, the term “orthorexia” has been given a new meaning. “Orthorexia”, in addition to the pathological dimension of orthorexia nervosa, may mean a non-pathological interest in healthy eating, called healthy orthorexia [25,44]. Our results show that there may be some association between celiac disease and the presence of healthy orthorexia in the study group. Patients at risk for orthorexia tended to pay less attention to the caloric content of their diet and snacked even more often than these without the risk. Such behavior may suggest that they did not focus on calorie counting, which is typical for some obsessive-compulsive spectrum disorders. On the other hand, they paid less attention to taste and more attention to food quality, felt confused upon entering the store and spent more time worrying about food. This may suggest their interest in eating foods appropriate for their condition, as a gluten-free diet can be challenging, especially for those who are shortly after diagnosis. Additionally, these individuals were less prone to mood swings and believed that eating healthy foods could improve appearance and change lifestyle. More research is needed to investigate whether the risk of orthorexia in individuals with celiac disease relates to a non-pathological or pathological interest in healthy eating.

In this survey, men were under-represented in the recruitment process and were ultimately excluded from the study. The reasons for this can be manifold. In celiac disease, women suffer more often from anemia, dyspepsia, constipation and genital disorders, and therefore, more often seek social media support in coping with these problems [45]. In contrast, men are generally less likely to participate in health promotion programs than women [46,47]. This results in a low recruitment rate, a delayed search for help and a reduced interest in health issues and habits. There are no previous studies considering healthy or unhealthy orthorexia in people with celiac disease. Although, in study of Ciacci et al. [48] women were significantly more likely than men to exhibit pathological healthy eating.

Strahler’s meta-analysis on sex differences in orthorexia suggests that further research on additional factors influencing pathological healthy eating should be pursued, thus our study attempts to find new risk factors for orthorexia among women [11]. While orthorexia is only slightly more prevalent in women than in men and no significant sex differences in the prevalence of orthorexia were found [49], our study suggests that a clinical disease such as celiac disease can affect eating behaviors in both non-pathological and pathological ways.

Although the GFD adherence questionnaire [50] was not used in the study, all of the females surveyed declared in the self-completed questionnaires that they strictly follow a gluten-free diet. In this context, allowing themselves to eating transgression can be interpreted differently, as eating unhealthy foods, snacking, eating irregularly and eating unhealthy foods, rather than not following a gluten-free diet. In comparison, in a study by Black et al. [51] as many as 96% of subjects with celiac disease strictly followed a gluten-free diet.

Another study observed that in individuals diagnosed with celiac disease, eating disorders and feelings of loss of gluten-containing meals were also associated with the desire to eat gluten-containing foods. In people without eating disorders who adapt well to celiac Disease diagnosis and do well with dietary self-management, there may be overly extreme concerns about cross-contamination. In others, the benefits of adhering to a GFD were not always perceived as beneficial, resulting in problems with dietary management [52]. The findings of our study and other researchers suggest that there is still a strong need for a powerful and reliable tool to assess specific eating behaviors in celiac disease.

The relationship between the presence of food neophobia and the presence of celiac disease was investigated in some studies, and it was found that in GFD users, celiac disease was the main determinant of food neophobia. Subjects with celiac disease had higher FNS score values (indicating greater food neophobia) than those using GFD, but without celiac disease [53]. As shown in the study by Sutherley et al. a useful tool for assessing food concerns in people with celiac disease in a clinical setting may be the Coeliac Disease Food Attitudes and Behaviours (CD-FAB). Cronbach’s alpha for CD-FAB was at a good level of 0.7, suggesting that it may be a reliable tool for measuring specific eating behaviors in celiac disease [54].

Another study used other research tools such as the quality of life (QOL) questionnaire and the Celiac Dietary Adherence Test (CDAT). 53% of individuals who were observed to have maladaptive eating behaviors toward maintaining a gluten-free diet, and were similar to known risk factors for eating and eating disorders, also had reduced QOL [55].

Adherence to a gluten-free diet is associated with a better quality of life for individuals with celiac disease [56]. In our study, following a gluten-free diet had positive impact on the self-assessed health status of people with celiac disease, which was better for a significant percentage of the participants.

The study by Wolf et al. [57] demonstrated potentially negative consequences of over-sensitivity in maintaining a strict gluten-free diet in individuals with celiac disease. Adults who were particularly hypersensitive to adhering to a gluten-free diet consistently ate at celiac-friendly restaurants, asked detailed questions at dinner and eliminated the potential for cross-contamination in their home cooking [57]. In studying a group of dietitians, Tremelling et al. [58] found that the risk that orthorexia may lie not only in healthy eating or obsessive food control, but also in increased preoccupation with body weight and body shape.

Our study found statistically significant differences in the practice of physical activity in females with and without risk of orthorexia. Those at risk for orthorexia were slightly most likely to engage in swimming while those without risk were most likely to engage in aerobics, strength training gym and running. Some studies have shown a positive link between orthorexia and physical activity [59,60,61,62,63]. No research was found on physical activity in people with celiac disease and co-occurring risk of orthorexia, but there are studies showing that the physical activity rates among females with celiac disease are at very low levels [64,65,66,67]. Engaging in low-impact or low-level physical activity may be associated with existing symptoms of celiac disease and may negatively affect the attitudes of people with celiac disease toward exercise and activity [68]. However, tailored exercise and lifestyle activities may lead to improvements in quality of life, exercise behaviors, and gastrointestinal symptoms in inactive adults with celiac disease and counteract developing additional chronic diseases [69,70]. More research is needed to expand knowledge about the relationship between exercise and orthorexia in people with celiac disease.

This study has several limitations. They are the lack of previous research on the prevalence of orthorexia in people with celiac disease, which precludes comparisons. In addition, our study included only females, making cross-gender comparisons impossible. The use of ORTO-15 questionnaire is a major limitation of the study. Stochel et al. [29] validated the ORTO-15 test in a group of 399 participants whose age was 15–21 years. Reliability analysis was performed based on the Cronbach’s α coefficient value and reached a satisfactory level (0.7–0.9). It was measured by the repeatability of the responses, the questionnaire presented very good (kappa: 0.81–1.00 for 5 items) and good repeatability (kappa: 0.61–0.80 for 10 items). The ORTO-15 questionnaire was found to be a reliable tool for identifying the risk of orthorexia in a population-based study in urban adolescents aged 15–21 years. Brytek-Matera et al. [30] recruited 341 females and 59 men (*n* = 400) whose ages ranged from 18 to 35 years. An internal consistency (Cronbach’s alpha) of 0.644 was demonstrated. The researchers performed exploratory and confirmatory factor analysis and the two-factor test structure was confirmed. It is well-known that people with celiac disease pay attention to their diet for health reasons. The Cronbach alpha is relatively low, which may reflect the limitations of the tool for use in this specific population. Due to the low Cronbach’s alpha coefficient, it is difficult to estimate the number of individuals at risk for orthorexia based on the ORTO-15 questionnaire alone and we cannot relate these results to the entire population of people with celiac disease. At present, there are several other research tools [71,72,73], but they had no Polish validated versions at the time we conducted our research, therefore, ORTO-15 was used in this study.

Another limitation was that the questionnaires on dietary behavior and physical activity were not validated in other studies.

The assessment of the reliability of this questionnaire, the use of an appropriate sample size of subjects, and the participation of subjects with clinically diagnosed celiac disease were the strengths of the study. Research on the relationship between orthorexia and celiac disease has not been performed before, so it was difficult to address similar data in the literature using comparative research methods. The ORTO-15 scale is only used to assess the risk of orthorexia, and not to make a full clinical diagnosis. A differential diagnosis should be carried out in order to assess whether the eating behaviors of persons with celiac disease are not due to food allergies or diseases requiring a special diet, as proposed by Moroze et al. [16].

We noted significant eating behaviors indicative of an interest in healthy food, such as feelings of guilt when transgressing the diet, feelings of confusion when entering a store, frequent thinking about food, paying attention to the calorie content of food, eating usually only three or four meals a day and believe that eating only healthy foods increased self-esteem. In general, new research tools are needed to distinguish pathological behavior from healthy orthorexia, which would facilitate assessment of high-risk groups.

## 5. Conclusions

The study was the first to look at the prevalence of orthorexia in adults with celiac disease. Given the survey instrument for assessing the prevalence of orthorexia and the overlap between eating behaviors in celiac disease and orthorexia, the prevalence of orthorexia in celiac disease cannot be clearly established. Future research should focus on using other research tools to confirm the presence of orthorexia in celiac disease.

## Figures and Tables

**Figure 1 nutrients-14-00904-f001:**
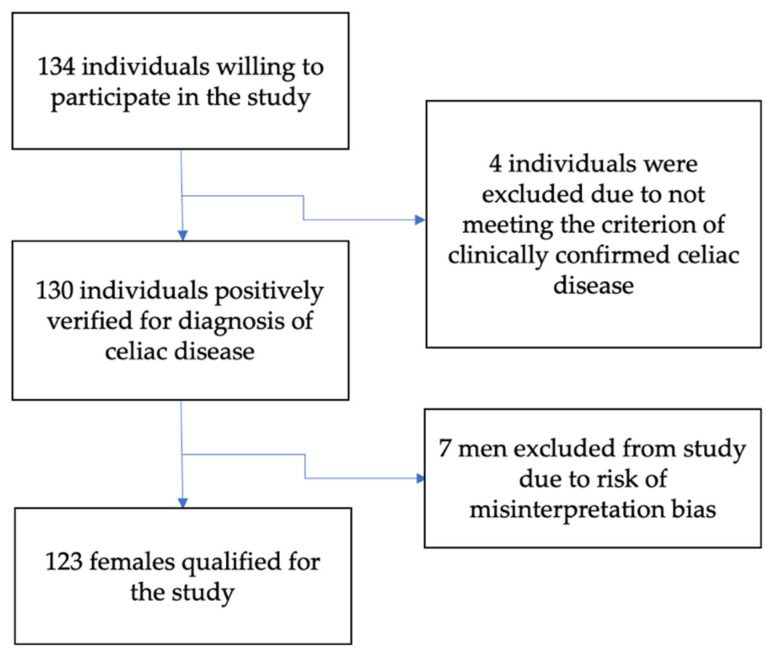
Flowchart of the study participants.

**Figure 2 nutrients-14-00904-f002:**
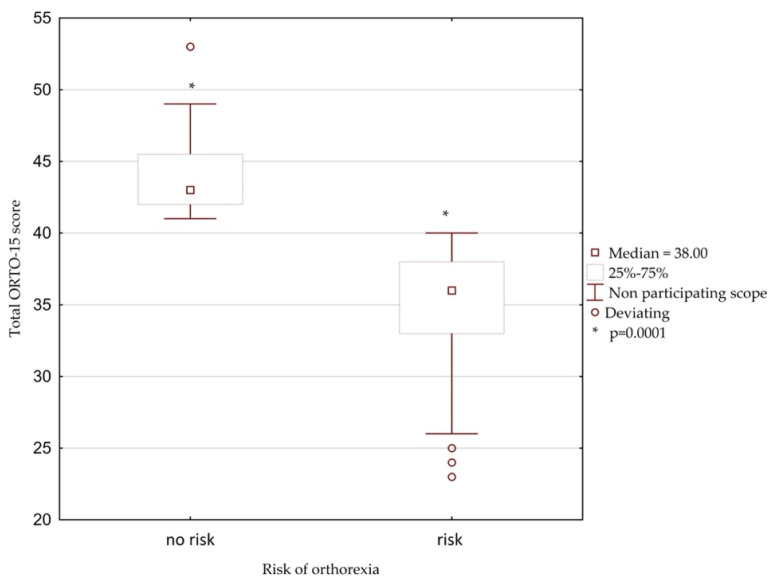
The box and mustache chart showing total ORTO–15 score in celiac participants with orthorexia.

**Figure 3 nutrients-14-00904-f003:**
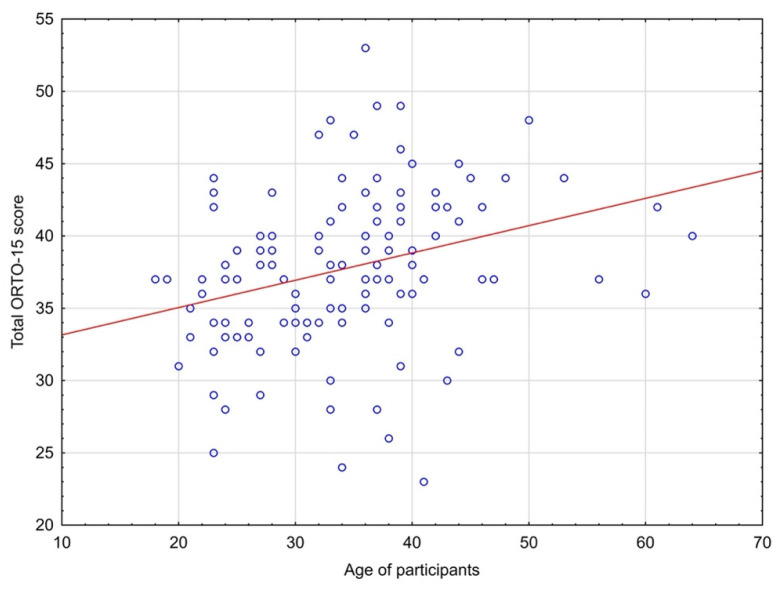
The relationship between age and total scores in the ORTO-15 test.

**Figure 4 nutrients-14-00904-f004:**
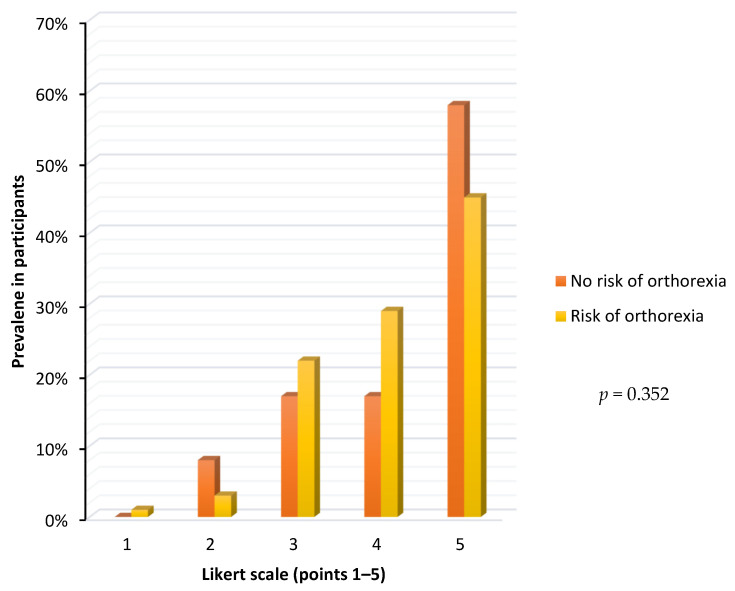
Self-assessed impact of celiac disease on the participants diet according to the orthorexia risk status.

**Table 1 nutrients-14-00904-t001:** General characteristics of the study group (*n* = 123).

Parameter	Value
Age (year)	
(Median/Q1–Q3)	34 (28–39)
(Average/age range/min-max)	34 ± 8.7/18–64
BMI (kg/m^2^)	
(Average/range)	21.51 (14.1–29.75)
(Median/Q1–Q3)	21.25 (19.1–23.6)
Education (%)	
Tertiary	80.5
Secondary	15.4
Middle	3.2
Primary	0.8
Occupation (%)	
Pension/retirement	1.6%
Odd job	1.6%
Unemployed	5.7%
Parental leave	8.1%
Students	8.1%
Permanent employment	74.8%
Weight Status (%)	
Underweight (moderate thinness) (16.0–16.99)	6%
Underweight (mild thinness) (17.0–18.49)	11%
Normal body weight (18.5–24.99)	73%
Overweight (25.0–29.99)	10%
Duration of celiac disease (n/%)	
Newly diagnosed (1–3 years)	29 (24%)
Patients with the celiac disease for at least 3 years	94 (76%)
Comorbidities (%)	
Lactose intolerance	23%
Hashimoto’s disease	22%
Hypothyroidism	17%
Food allergy	14%
Duhring’s disease	6.5%
The most common symptoms reported before treatment of celiac patients	
Bloating	53%
Chronic diarrhea	49%
Anemia	41%
Low body weight	24%
Adherence to gluten-free diet (%)	
Full	100%
Health self-assessment on a gluten-free diet (%)	
Better	87%
No change	10%
Worse	3%

**Table 2 nutrients-14-00904-t002:** Reliability analysis of individual questions of the ORTO-15 questionnaire.

Questions from ORTO-15 Test	Score * (Mean ± SD)	Spearman’s Rho	Cronbach’s Alpha Coefficient **	Mean ± SD of Total Points ***
1. When eating, do you pay attention to the calories of the food?	2.91 ± 0.90	0.33	0.66	34.82 ± 5.07
2. When you go in a food shop do you feel confused?	3.34 ± 0.89	0.31	0.66	34.39 ± 5.08
3. In the last 3 months, did the thought of food worry you?	2.57 ± 1.00	0.53	0.63	35.16 ± 4.83
4. Are your eating choices conditioned by your worry about your health status?	1.66 ± 0.61	0.15	0.68	36.07 ± 5.30
5. Is taste of food more important than the quality when you evaluate food?	2.53 ± 0.77	0.12	0.68	35.20 ± 5.29
6. Are you willing to spend more money to have healthier food?	2.01 ± 0.66	0.11	0.68	35.72 ± 5.32
7. Does the thought about food worry you for more than three hours a day?	3.34 ± 0.87	0.55	0.63	34.38 ± 4.90
8. Do you allow yourself any eating transgressions?	3.00 ± 0.82	−0.03	0.70	34.73 ± 5.39
9. Do you think your mood affects your eating behavior?	2.33 ± 0.79	0.58	0.63	35.43 ± 4.93
10. Do you think that the conviction to eat only healthy food increases self-esteem?	2.96 ± 0.93	0.51	0.63	34.77 ± 4.90
11. Do you think that eating healthy food changes your life-style (frequency of eating out, friends…)?	2.38 ± 0.92	0.34	0.66	35.35 ± 5.05
12. Do you think that consuming healthy food may improve your appearance?	2.07 ± 0.82	0.38	0.65	35.67 ± 5.06
13. Do you feel guilty when transgressing?	2.52 ± 1.11	0.26	0.67	35.21 ± 5.03
14. Do you think that on the market there is also unhealthy food?	1.43 ± 0.68	0.02	0.69	36.30 ± 5.35
15. At present, are you alone when having meals?	2.72 ± 0.83	0.09	0.69	35.02 ± 5.29

* Score of 4-point scale (always, often, rarely or never) of ORTO-15; ** Cronbach’s alpha coefficient after removal of individual questions; *** ORTO-15 questionnaire after excluding an individual question.

**Table 3 nutrients-14-00904-t003:** Reliability analysis for the ORTO-15 questionnaire overall.

Reliability Analysis	Mean ± SD *	Min-Max *	Cronbach’s Raw α Coefficient	Standardized Cronbach’s α Coefficient	Average Spearman’s Rho between Questions
ORTO-15 questionnaire (15 scale items)	37.73 ± 5.45	23–53	0.67	0.66	0.11
ORTO-15 questionnaire (excluding question 8)	34.73 ± 5.41	20–50	0.70	0.68	0.13

* SD–standard deviation; Min-Max–Minimum-Maximum.

**Table 4 nutrients-14-00904-t004:** Assessment of the prevalence of orthorexia risk based on ORTHO-15 test scores.

Risk of Orthorexia	ORTHO-15 *	
	*n* *	%
The entire study group	123	100
Participants at risk of orthorexia **	87	71
Participants at risk of orthorexia ***	39	32

* *n* = number of respondents; ORTO-15 risk assessment of orthorexia with a 40-item cut-off on the ORTO-15 questionnaire; ** cut-off score of 40 points; *** cut-off score of 35 points.

**Table 5 nutrients-14-00904-t005:** The differences between eating habits and behaviors and the incidence of orthorexia risk.

Eating Habits and Behaviors *		Risk of Orthorexia		χ² Test *p*
		Yes (*n* = 87)	No (*n* = 36)	
Frequency of meals per day	3 times a day	36%(*n* = 31)	19% (*n* = 7)	*p* = 0.014
	4 times a day	38% (*n* = 33)	67% (*n* = 24)	
	5 times a day	26% (*n* = 23)	14% (*n* = 5)	
Independent preparation of gluten-free meals		94% (*n* = 82)	78% (*n* = 28)	*p* = 0.006
Difficulty eating out		78% (*n* = 68)	92%% (*n* = 33)	*p* = 0.075
Paying attention to the caloric content of gluten-free meals		54% (*n* = 47)	53% (*n* = 19)	*p* = 0.899
Paying attention to the composition of gluten-free products		87% (*n* = 76)	92% (*n* = 33)	*p* = 0.493
Physical activity *				
Total number of active people		70% (*n* = 61)	81% (*n* = 29)	*p* = 0.234
Aerobic (fitness)		13% (*n* = 13)	31% (*n* = 11)	*p* = 0.046
Swimming		9% (*n* = 8)	6% (*n* = 2)	*p* = 0.050
Strength training gym		9% (*n* = 8)	22% (*n* = 8)	*p* = 0.050
Running		5% (*n* = 4)	17% (*n* = 6)	*p* = 0.025
Walking		46% (*n* = 40)	53% (*n* = 19)	*p* = 0.492
Yoga		5% (*n* = 4)	0%	*p* = 0.190
Cycling		30% (*n* = 26)	42% (*n* = 15)	*p* = 0.207
Frequency of physical activity		71% (*n* = 87)	29% (*n* = 36)	*p* = 0.599
1 time per week		16% (*n* = 14)	11% (*n* = 4)	
2–3 times a week		37% (*n* = 32)	44% (*n* = 16)	
4 and more per week		14% (*n* = 12)	17% (*n* = 6)	
Daily		9% (*n* = 8)	14% (*n* = 5)	
No physical activity		24% (*n* = 21)	14% (*n* = 5)	

* multiple-response question.

**Table 6 nutrients-14-00904-t006:** Differences between the risk group of orthorexia and non-risk participants in ORTO-15 questionnaire.

	Risk of Orthorexia		χ² Test
ORTO-15 Questions	Yes (*n* = 87)	No (*n* = 36)	*p*
1. When eating, do you pay attention to the calories of the food?			*p* = 0.001
Always/Often %	46%**	69%	
2. When you go in a food shop do you feel confused?			*p* = 0.003
Always/Often %	73%	94%	
3. In the last 3 months, did the thought of food worry you?			*p* = 0.001
Always/Often %	64%	8%	
4. Are your eating choices conditioned by your worry about your health status?			*p* = 0.086
Always/Often %	92%	95%	
5. Is taste of food more important than the quality when you evaluate food?			*p* = 0.043
Always/Often %	49%	72%	
6. Are you willing to spend more money to have healthier food?			*p* = 0.071
Always/Often %	85%	67%	
7. Does the thought about food worry you for more than three hours a day?			*p* = 0.001
Always/Often %	22%	0%	
8. Do you allow yourself any eating transgressions?			*p* = 0.437
Always/Often %	73%	87%	
9. Do you think your mood affects your eating behavior?			*p* = 0.001
Always/Often %	20%	80%	
10. Do you think that the conviction to eat only healthy food increases self-esteem?			*p* = 0.001
Always/Often %	43%	6%	
11. Do you think that eating healthy food changes your life-style (frequency of eating out, friends…)?			*p* = 0.001
Always/Often %	68%	31%	
12. Do you think that consuming healthy food may improve your appearance?			*p* = 0.001
Always/Often %	83%	59%	
13. Do you feel guilty when transgressing?			*p* = 0.001
Always/Often %	42%	39%	
14. Do you think that on the market there is also unhealthy food?			
Always/Often %	95%	94%	*p* = 0.012
15. At present, are you alone when having meals?			
Always/Often %	41%	42%	*p* = 0.819

## Data Availability

Not applicable.

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
