# Peer review of "Dietary Behavior and Risk of Orthorexia in Women with Celiac Disease"

_nutrients, 2022, doi:10.3390/nu14040904_

Round 1
Reviewer 1 Report
I read with great interest the following manuscript: Dietary behavior and risk of orthorexia in women with celiac disease. This manuscript attempts to add to the increasing knowledge on the relationships between disordered eating, eating disorders and a diagnosis of celiac disease and this is clearly detailed in the introduction. Interrelationships between orthorexia nervosa and celiac disease have not previously been explored, a novel aspect of this manuscript. However, I have several concerns that require further clarification, to strengthen this manuscript:
- The ORTO-15 has not been validated for use in a celiac disease population, and as such, the authors assess Cronbach’s alpha to understand internal consistency within this population. The Cronbach’s alpha was low (0.66) meaning some items were not representative of the behavior in this population. As such, item 8 was removed. Despite removal of item 8, the cut-off values remained consistent with the 15-item version of the ORTO (effectively making it harder to score above 35 or 40). Please provide a rationale for keeping the original cut-offs, or revise.
- It is perhaps unsurprising that the 15 item ORTO has poor internal consistency within this population – as high scores on several items (e.g., when you go in a food shop do you feel confused?) may be indicative of effective management of the gluten-free diet. This has been highlighted in a number of articles assessing prevalence of disordered eating in celiac disease, with suggestions to develop coeliac-disease specific measures. In fact, some measures have already been developed (the celiac disease food attitudes and behaviours scale) – it would be helpful to reflect on these measures within your discussion.
- Please provide a sample size calculation or power analysis for your study.
- Figure 4 – the labels for the x and y axis need to be clearly labelled.
- The structure of the discussion could do with some reworking – it currently has a lot of overlap with the results.
- Key references that are missing, some of which have been published by this journal include:
- Cadenhead et al (2019). Diminished quality of life among adolescents with coeliac disease using maladaptive eating behaviours to manage a gluten-free diet: a cross-sectional, mixed-methods study. DOI: 10.1111/jhn.12638
- Zysk et al (2019). Food Neophobia in Celiac Disease and Other Gluten-Free Diet Individuals. doi: 10.3390/nu11081762
- Satherley et al (2018). Development and Validation of the Coeliac Disease Food Attitudes and Behaviours Scale. DOI: 10.1155/2018/6930269
Author Response
Response to Reviewer 1 Comments
Reviewer 1 comments
I read with great interest the following manuscript: Dietary behavior and risk of orthorexia in women with celiac disease. This manuscript attempts to add to the increasing knowledge on the relationships between disordered eating, eating disorders and a diagnosis of celiac disease and this is clearly detailed in the introduction. Interrelationships between orthorexia nervosa and celiac disease have not previously been explored, a novel aspect of this manuscript. However, I have several concerns that require further clarification, to strengthen this manuscript:
Response: Dear Reviewer! Thank you for your valuable tips and hints. We have read all the proposals carefully and have responded to each one. We hope that we have solved the problem as well as possible.
Point 1: The ORTO-15 has not been validated for use in a celiac disease population, and as such, the authors assess Cronbach’s alpha to understand internal consistency within this population. The Cronbach’s alpha was low (0.66) meaning some items were not representative of the behavior in this population. As such, item 8 was removed. Despite removal of item 8, the cut-off values remained consistent with the 15-item version of the ORTO (effectively making it harder to score above 35 or 40). Please provide a rationale for keeping the original cut-offs, or revise.
Response 1: In line 150 we added that the questionnaire for coeliac patients was not validated and therefore evaluated the reliability of the questionnaire.
We decided to keep the original 40 points in our study, as our study was the first of its kind to use the ORTO-15 questionnaire for celiac disease. Therefore, we kept the original ORTHO-15 criteria in order to compare our research results with other studies. This information has been added in the methodology.
Point 2: It is perhaps unsurprising that the 15 item ORTO has poor internal consistency within this population – as high scores on several items (e.g., when you go in a food shop do you feel confused?) may be indicative of effective management of the gluten-free diet. This has been highlighted in a number of articles assessing prevalence of disordered eating in celiac disease, with suggestions to develop coeliac-disease specific measures.
Response 2: We raised this point in the debate and said that the instrument should be constantly developed in order to increase credibility and sensitivity in relation to celiac disease.
Point 3: In fact, some measures have already been developed (the celiac disease food attitudes and behaviours scale) – it would be helpful to reflect on these measures within your discussion.
Response 3: The discussion included information on the development and validation of the Coeliac Disease Food Attitudes and Behaviours Scale (Sutherley), the Quality of Life Questionnaire (QOL), the Celiac Dietary Adherence Test (CDAT) and the GFD adherence questionnaire.
Point 4: Please provide a sample size calculation or power analysis for your study.
Response 4: In the study, we have given the calculations of the sample size under the original assumptions. “The sample size was calculated on the basis of the prevalence of coeliac disease in the Polish population, which is estimated at 1% with a 95% confidence interval and a margin of error of 1,71%.” We have added this information to the materials and methodology of the study group.
Point 5: Figure 4 – the labels for the x and y axis need to be clearly labelled.
Response 5: Thank you for using the x- and y-axis labels of Figure 4. We decided to change labels on Figure 4 because we think they can improve readability of this figure.
Point 6: The structure of the discussion could do with some reworking – it currently has a lot of overlap with the results.
Response 6: We have deleted some of the information that was similar to the results, in particular the information on the reliability of the questionnaire at the beginning. However, when we refer to our research findings in our discussion, we discuss the subject and the problems of the subject and refer to other work.
Point 7: Key references that are missing, some of which have been published by this journal include:
- Cadenhead et al (2019). Diminished quality of life among adolescents with coeliac disease using maladaptive eating behaviours to manage a gluten-free diet: a cross-sectional, mixed-methods study. DOI: 10.1111/jhn.12638
- Zysk et al (2019). Food Neophobia in Celiac Disease and Other Gluten-Free Diet Individuals. doi: 10.3390/nu11081762
- Satherley et al (2018). Development and Validation of the Coeliac Disease Food Attitudes and Behaviours Scale. DOI: 10.1155/2018/6930269
Response 7: We have familiarized ourselves with the references mentioned. We believe that there are interesting researches, worthy of attention that provides valuable information. We decided to include that references to the discussion.

Reviewer 2 Report
In this study, the Authors analyse the prevalence of risk factor for orthorexia in celiac disease patients. It is an interesting subject of research and I find there is a need for data in this field.
I have some issues/questions regarding the study:
1. The Authors report that 100% of the participants followed a strict gluten free diet: first of all, no GFD Adherence questionnaire was administered (like the Corazza questionnaire) to corroborate this data. Second, the explanations for this selection bias and its possible influence on the results are not discussed.
2. Similar to above, the fact that only a handful of men responded to recruitment can't solely be explained with the higher prevalence of celiac disease in women, but is not discussed further in the Discussion.
3. A more thorough discussion about the applicability of this Questionnaire in the particular field of celiac disease is needed. For example, as gastroenterologist working in a celiac disease clinic I find specific issues with Questions 2-4-6-8-13 that can be interpreted differently by people following a specific, life-long diet. Would it be possible to include a new analysis and critical interpretation to see the specific weight of this group of questions on the final scores?
4. Along with the correlation between age and score, a correlation between "age of disease" (years since diagnosis) and score should be performed because it would really help to see if anxiety and difficulties due to CD diagnosis and early diet years play a role in the ORTHO-tex score.
5. I seem to understand that the prevalence of "risk for orthorexia in celiac patients" was not much higher than in the healthy population (for example Uni students), or do the Authors es expert judge it is a high prevalence? Since there was no matched control group in this study (a pity in my opinion because it would have helped to evaluate the reliability of data), a conclusive "expert opinion" on the assessment of how high the risk for orthorexia in celiac patients is should be put on foreground.
Other minor issues:
Introduction, Page 2, line 66 please explain the association between orthorexia and "gastritis". Gastritis is an endoscopic/histologic diagnosis which can, among others, be at the basis of restrictive diets. Is there a proved association between orthorexia and gastritis? I suggest to add the proper citation or to substitute the term with dyspepsia or epigastric pain.
Introduction, Page 2, lIne 74-75: The sentence "Passannanti et al. [24] found that celiac disease itself, rather 74 than gastrointestinal symptoms or psychological factors, may contribute to pathological 75 eating behaviors in adults with celiac disease." is not clear, please reformulate or explain.
Introduction, Line 81: "With the increasing trend of celiac disease, " is also unclear, do you mean the increasing prevalence due to improved diagnosis? Please specify.
Methods, Line 117: "those who had been ill for more than 3 years" if find this description inappropriate when it comes to celiac disease, because in the field of celiac disease it leads to think that people in this group have had symptoms for longer than 3 years, while people with celiac disease on a gluten free diet usually have no symptoms.
Results Line 164: it was said before that males were excluded and only females were included in the study group, moreover the Table 1 reports all results about the 123 females, so I thin the sentence "The study group was predominantly females (n=123, approximately 95%)" is incorrect. The study group was 100% females (95% of the initially recruited before the exclusion of males?).
Author Response
Response to Reviewer 2 Comments
Reviewer 2 comments
In this study, the Authors analyse the prevalence of risk factor for orthorexia in celiac disease patients. It is an interesting subject of research and I find there is a need for data in this field.
Response: Dear Reviewer! Thank you for your valuable tips and hints. We have read all the proposals carefully and have responded to each one. We hope that we have solved the problem as well as possible.
I have some issues/questions regarding the study:
Point 1: The Authors report that 100% of the participants followed a strict gluten free diet: first of all, no GFD Adherence questionnaire was administered (like the Corazza questionnaire) to corroborate this data. Second, the explanations for this selection bias and its possible influence on the results are not discussed.
Response 1: Thank you for pointing this out. Unfortunately, it is no longer possible to include the GFD Adherence Questionnaire in the research work, but we have included in the discussion information that a strict diet among the participants was not determined on the basis of a validated questionnaire, e.g. GFD Adherence, or on the basis of our questions, which might have an impact on the research results.
Point 2: Similar to above, the fact that only a handful of men responded to recruitment can't solely be explained with the higher prevalence of celiac disease in women, but is not discussed further in the Discussion.
Response 2: When planning our study, we assumed that we would attract a correspondingly large group of men and women. Unfortunately, we were unable to bring together a sufficiently large group of men. We also knew that the incidence of celiac disease in Poland is higher among women than among men (2:1 ratio; female:male ratio). In another study, gastrointestinal symptoms occur more frequently in women than in men. Women’s willingness to participate in research may therefore be greater and may be linked to the need to seek support, help and examination of complaints. We have decided to include this information in our work.
Point 3: A more thorough discussion about the applicability of this Questionnaire in the particular field of celiac disease is needed. For example, as gastroenterologist working in a celiac disease clinic I find specific issues with Questions 2-4-6-8-13 that can be interpreted differently by people following a specific, life-long diet. Would it be possible to include a new analysis and critical interpretation to see the specific weight of this group of questions on the final scores?
Response 3: In the reliability analysis we considered the understanding of weighting only for questions 2-4-6.8.13. We summed up the importance of that questions and found the results:
Mean: 12.52 ± 1.88; Cronbachs alpha: 0.010 Standardized alpha:0.04; Correlation: 0.011. Analysis of these questions alone has greatly reduced the reliability of Cronbach’s alpha, suggesting the specificity of questions 2-4-6.8.13. However, it is difficult to interpret the final result without the other 8 questions.
We performed another integrity analysis and excluded questions 2-4-6.8.13 from the test and obtained these results: Mean: 25.20 ± 4.35; Min. 14 and max. 37; Cronbachs alpha: 0.66 Standardized alpha:0.67; Correlation: 0.17. In our view, not only Question 8 affects the reliability of the questionnaire, but also Question 2-4-6.8.13. These questions may affect the overall ORTO-15 result and may be related to a specific group of coeliac patients.
Point 4: Along with the correlation between age and score, a correlation between "age of disease" (years since diagnosis) and score should be performed because it would really help to see if anxiety and difficulties due to CD diagnosis and early diet years play a role in the ORTHO-tex score.
Response 4: Thank you for your valuable advice. We examined the correlation between age of disease (years since diagnosis) and outcome (ORTO-15 sum). At a significance level of alpha<0.05 (95% confidence interval), a critical value of p=0.20 with a weak negative correlation of -0.115 was obtained. The mean for years of diagnosis was 2011 with a standard deviation of 9.60. Using a chi-square test, we examined the relationship of the age of illness divided into: up to 3 years and more than 3 years in the context of the risk of orthorexia. At a 95% confidence level, the critical value of the test was p=0.50, indicating no association between the risk of orthorexia and the age of the disease. After coding the years of diagnosis into 1- to 3 years and 2 over three years and performing a correlation with the ORTO-15 summary score, the critical value of the correlation was p=0.943. In view of the above results, we decided not to include this information in the paper.
Point 5: Other minor issues:
Introduction, Page 2, line 66 please explain the association between orthorexia and "gastritis". Gastritis is an endoscopic/histologic diagnosis which can, among others, be at the basis of restrictive diets. Is there a proved association between orthorexia and gastritis? I suggest to add the proper citation or to substitute the term with dyspepsia or epigastric pain.
Response 5: Thanks for the tip on gastritis in orthorexia. We assume that there may be a link between functionally induced orthorexia, dyspepsia or abdominal pain, but there are no detailed studies on the link between gastritis and orthorexia. There are reports of a positive relationship between healthy eating and progress in treating gastritis (Martina Valente, Elena V. Syurina, Seda Muftugil-Yalcin & Tomris Cesuroglu (2020) “Keep Yourself Alive”: From Healthy Eating to Progression to Orthorexia Nervosa A Mixed Methods Study among Women in the Netherlands, Ecology of Food and Nutrition, 59:6, 578-597, DOI: 10.1080/03 670 244.2020.1 755 279. That’s why we decided to change the term “gastritis” to “pain in the upper abdomen.”
Point 6: Introduction, Page 2, lIne 74-75: The sentence "Passannanti et al. [24] found that celiac disease itself, rather 74 than gastrointestinal symptoms or psychological factors, may contribute to pathological 75 eating behaviors in adults with celiac disease." is not clear, please reformulate or explain.
Response 6: We have reformulated an unclear sentence on the basis of the Passannanti study. First, a distinction should be made between celiac disease without gastrointestinal symptoms and gastrointestinal symptoms without accompanying celiac disease.
Passannanti et al. found that the occurrence of celiac disease as a disease contributes to pathological eating, regardless of whether it is associated with gastrointestinal diseases, which may also indicate the psychological aspect of the onset of the disease, and that gastrointestinal symptoms without celiac disease cannot contribute to unfavourable eating behaviour.
Point 7: Introduction, Line 81: "With the increasing trend of celiac disease, " is also unclear, do you mean the increasing prevalence due to improved diagnosis? Please specify.
Response 7: Thank you for your special attention to the section on the tendency of celiac disease. In our work, we have explained what can be associated with the increase in coeliac disease. For example, in a study in Denver, CO, USA, children with an increased genetic risk of celiac disease were observed over a period of 20 years. When these results were extrapolated to the total population of the city, the cumulative incidence of coeliac disease was found to be 1-6% at the age of 5 years, 2-8% at the age of 10 years and 3-1% at the age of 15 years, which is a remarkable incidence rate similar to that of the Scandinavians (Liu E, Dong F). Barón AE, Barón AE, et al. Celiac disease in a long-term study of adults with genotyping. Gastroenterology 2017; 152: 1329-36.). Some studies suggest an increase in celiac disease due to more frequent diagnoses and some suggest a higher incidence of celiac disease.
Point 8: Methods, Line 117: "those who had been ill for more than 3 years" if find this description inappropriate when it comes to celiac disease, because in the field of celiac disease it leads to think that people in this group have had symptoms for longer than 3 years, while people with celiac disease on a gluten free diet usually have no symptoms.
Response 8: Thank you for your valuable advice. The phrase “those who had been ill for more than 3 years” might have been misleading and indicate symptoms of celiac disease, not the disease itself. Therefore, we changed it to “have been diagnosed with celiac disease for more than 3 years.”
Point 9: Results Line 164: it was said before that males were excluded and only females were included in the study group, moreover the Table 1 reports all results about the 123 females, so I thin the sentence "The study group was predominantly females (n=123, approximately 95%)" is incorrect. The study group was 100% females (95% of the initially recruited before the exclusion of males?).
Response 9: Thank you for pointing this out. The sentence “The study group consisted mainly of women (n=123, about 95%) “ may have confused the reader. Therefore, we have corrected this section in the paper and added the information that the study group consisted 100% of women.
